METHODS

# Krisp: A Python package to aid in the design of CRISPR and amplification-based diagnostic assays from whole genome sequencing data

**Zachary S. L. Foster, Andrew S. Tupper, Caroline M. Press, Niklaus J. Grünwald** *

Horticultural Crops Disease and Pest Management Research Unit, USDA Agricultural Research Service, Corvallis, Oregon, United States of America

* nik.grunwald@usda.gov

**Data Availability Statement:** All krisp package is available at https://github.com/grunwaldlab/krisp and https://pypi.org/project/krisp/.

## Abstract

Recent pandemics like COVID-19 highlighted the importance of rapidly developing diagnostics to detect evolving pathogens. CRISPR-Cas technology has recently been used to develop diagnostic assays for sequence-specific recognition of DNA or RNA. These assays have similar sensitivity to the gold standard qPCR but can be deployed as easy to use and inexpensive test strips. However, the discovery of diagnostic regions of a genome flanked by conserved regions where primers can be designed requires extensive bioinformatic analyses of genome sequences. We developed the Python package `krisp` to aid in the discovery of primers and diagnostic sequences that differentiate groups of samples from each other, using either unaligned genome sequences or a variant call format (VCF) file as input. `Krisp` has been optimized to handle large datasets by using efficient algorithms that run in near linear time, use minimal RAM, and leverage parallel processing when available. The validity of `krisp` results has been demonstrated in the laboratory with the successful design of a CRISPR diagnostic assay to distinguish the sudden oak death pathogen *Phytophthora ramorum* from closely related *Phytophthora* species. `Krisp` is released open source under a permissive license with all the documentation needed to quickly design CRISPR-Cas diagnostic assays.

## Author summary

Pathogens continue to emerge at accelerated rates affecting animals, plants, and ecosystems. Rapid development of novel diagnostic tools is needed to monitor novel pathogen variants or groups. We developed the computational tool `krisp` to identify genetic regions suitable for development of CRISPR diagnostics and traditional amplification-based diagnostics such as PCR. `Krisp` scans whole genome sequence data for target and non-target groups to identify diagnostic regions based on DNA or RNA sequences. This computational tool has been validated using genome data for the sudden oak death pathogen *Phytophthora ramorum*. `Krisp` is released open source under a permissive license with all the documentation needed to quickly design CRISPR-Cas diagnostic assays and other amplification-based assays.

**Funding:** NJG was funded by USDA ARS (2072-22000-041-000-D) and USDA NIFA (2023-67013-39918). CMP received salary from 2072-22000-041-000-D. ZSLF received salary from 2072-22000-041-000-D and 2023-67013-39918. AST received salary from 2023-67013-39918. The funders had no role in study design, data collection and analysis, decision to publish, or preparation of the manuscript.

## Introduction

Invasive organisms continue to emerge at accelerated rates worldwide, likely due to increased global trade, movement of live plants, a growing human population, and stresses imposed by the Anthropocene on hosts [1–5]. Most pathogens, whether bacteria, fungi, oomycetes, viruses, nematodes, or other protists, are hard to detect without appropriate diagnostics. The biosurveillance of the future will require affordable, robust, accurate, and field deployable diagnostic methods that can be developed quickly. The capacity to develop novel diagnostics in response to an emergent pathogen in weeks, rather than months or years, would be transformative. Whole genome sequences (WGS) of new variants combined with computational tools to identify diagnostic loci provides a new means of accelerating developing diagnostics in response to emerging or reemerging invasive pathogens and pests.

Clustered regularly interspaced short palindromic repeats (CRISPR) are DNA sequences found in prokaryotes to detect and counter viral infections [6]. These loci consist of a series of conserved repeats alternating with equal length variable sequences referred to as spacers. The spacers are sequences copied from viruses that the microbe or its progenitors have encountered in the past. CRISPR-associated proteins (Cas) use RNA transcribed from the spacer sequences as guides to recognize and, depending on the CAS enzyme, degrade specific strands of DNA or RNA. For example, the Cas9 protein introduces double-stranded breaks near DNA sequences complementary to the bound guide RNA (Jinek et al., 2012). In addition to sequence-specific cleavage, the Cas12, Cas13, and Cas14 families of proteins cause cleavage of off-target nucleic acid polymers after binding to and cleaving a target sequence, a phenomena referred to as collateral cleavage [7–10]. For Cas13 orthologs, both sequence-specific cleavage and collateral cleavage target RNA [11], whereas Cas12 and the smaller Cas14 variant target ssDNA or dsDNA and cleave bystander ssDNA [8,10]. Cas enzymes also differ in regard to which 2-6bp sequence motifs are most effected by collateral cleavage [11]. As part of the bacterial immune system, collateral cleavage is thought to limit pathogen propagation via programmed cell death or dormancy induction [7], thereby sacrificing the individual to benefit the population as a whole.

The discovery and understanding of the CRIPSR-Cas immune system has led to many practical advances in molecular biology, including CRISPR-based diagnostic assays for detecting specific RNA or DNA sequences (CRISPR-dx). Many techniques rely on using the collateral cleavage activity of the Cas12, Cas13, and Cas14 enzymes to degrade ssDNA or RNA reporters as a measurable signal of sequence recognition. The degradation of reporter molecules can be detected by a variety of ways, including measuring fluorescence, imaging fluorescent bands on a gel, or observing bands on a lateral flow device [12]. In the case of solution-based fluorometric detection, short off-target RNA or ssDNA polymers, each with a fluorescent probe on one end and a quencher on the other, are cut by the collateral cleavage of Cas proteins upon sequence recognition, resulting in fluorescence [9]. This general technique has been used to create highly specific CRISPR-based diagnostic assays. SHERLOCK [12] and DETECTR [8] achieve high sensitivity using amplification of target DNA by recombinase polymerase amplification (RPA), an isothermal alternative to PCR [13]. HOLMES [14] uses loop-mediated isothermal amplification (LAMP), another isothermal amplification technique [15]. CONAN achieves high sensitivity without preamplification by using a sophisticated DNA/RNA hybrid reporter that, when degraded, releases a guide RNA that matches a fragment of DNA added to the reaction, resulting in exponential activation of the Cas12a enzyme [16]. These assays have been shown to be both highly specific, allowing single nucleotide discrimination, and very sensitive, with detection thresholds in the attomolar range [8,11]. Sensitivity can be further increased by scaling up preamplification or adding multiple guide RNAs targeting different

parts of the same target sequence into the same reaction [17]. These methods can be adapted to single-tube isothermal reactions or lateral flow strips, allowing for on-site detection of pathogens with minimal specialized instruments or training for as little as $0.60 per assay [12]. Other CRISPR-dx techniques involve engineered tracrRNAs to make the target RNA act as a guide RNA (LEOPARD) [18], observing liquid-liquid phase separation caused by collateral cleavage [19], engineered hydrogels that change their material properties in response to a target [20], measuring the byproducts of the polymerase activity of Cas10 [21], using two guide RNAs to initiate strand displacement amplification (CRISDA) [22], and the electronic measurement of genomic sequence binding to graphene field-effect transistors [23]. Although still a new technology, there are already CRISPR-based tests approved by the United States Food and Drug Administration to target SARS-CoV-2 [24].

CRISPR-dx assays can generally be designed to target any sequence, but the criteria for designing optimal guide RNAs depend on the specific method and Cas enzyme used. Cas enzymes differ in their requirements for adjacent sequence motifs, the length of guide RNA required, and how mismatches between the guide RNA and target affect cleavage efficiency. For example, Cas12 enzymes target both ssDNA and dsDNA, but for dsDNA targets, a protospacer adjacent motif (PAM) (e.g. TTTV) must be present in the target sequence near where the guide RNA binds [12]. This requirement can restrict which portions of the genome can be targeted, unless the PAM sequence is added to the 5' end of one of the primers during a preamplification step [14]. The optimal length of the guide RNA can be different for each Cas ortholog: Cas13a uses a 28nt guide RNA, Cas13b uses a 30nt guide RNA, and Cas12a uses a 20nt guide RNA [12]. Finally, how much of an effect a mismatch between the guide RNA and the target has depends on both the location of the mismatch and the Cas enzyme used. For example, Cas13a is more sensitive to mismatches at the 3rd position, whereas Cas12b is more sensitive to mismatches in the 10th to 16th positions [14].

In addition to a specialized guide RNA, many methods require designing primers for preamplification of target DNA to increase sensitivity enough for the method to be useful for detecting pathogens in clinical settings. Amplification of target DNA is often done using an isothermal technique such as RPA or LAMP to minimize the need for expensive equipment and make it easier to conduct tests outside of laboratories [24]. Various non-target sequences may have to be added to the primer sequences, such as the previously mentioned PAM sequence required by Cas12 when targeting dsDNA [14]. Cas13 orthologs detect and cleave RNA, so to detect a DNA target a T7 promoter must be added to the 5' end of one of the primers used for preamplification to allow for transcription [12]. These modifications can affect how primers and loci are chosen when designing a CRISPR-dx diagnostic assay.

Development of CRISPR-dx assays typically require the design of a diagnostic CRISPR guide RNA (crRNA) that discriminates between target and non-target sequences and sequence-specific primers for amplification of the target region. The abundance of public WGS data, combined with the affordability of generating new sequences for novel variants, has the potential to make designing new diagnostic assays fast and reliable. However, finding optimal candidate regions can be difficult when the assay is applied to organisms with large genomes or populations with high sequence diversity. Although many tools exist to discover crRNAs for genetic engineering, few exist that identify the crRNA and associated primers needed for CRISPR-dx. Current approaches either only work on some model organisms [25], do not incorporate primer design [26], or are designed towards specific types of CRISPR-dx assays [27]. We developed the Python package `krisp` to identify candidate regions and primers for development of CRISPR-dx assays as well as any other amplification-based assays requiring primers flanking a diagnostic region, such as PCR `krisp` takes as input either a set of unaligned FASTA files representing assembled genomes or a Variant Call Format (VCF) file

containing variants called against a reference genome. `krisp` has been designed to run in parallel, use minimal RAM, and have run times that correlate linearly with input size, allowing for the comparison of hundreds to thousands of genomes. For FASTA input, `krisp` breaks sequences into k-mers and applies sequential filtering steps to find diagnostic regions that distinguish a target group from all outgroups and are flanked by conserved regions where primers can be designed. For VCF input, `krisp` identifies clusters of diagnostic variants flanked by regions without variants and infers the sequence for each group by applying variants to the reference sequence. For both input types, primers can be designed automatically using Primer3 [28] and the candidate regions can be filtered based on the presence and quality of possible primers. Candidate regions are reported as either human readable alignments or tabular data, allowing the user to apply further processing steps in a custom pipeline. Finally, `krisp` is highly flexible and can be configured to search for regions and primers compatible with any DNA/RNA probe/primer diagnostics as well as specific CRISPR-dx assays, such as SHER-LOCK or DETECTR.

## Materials and methods

The `krisp` Python package has two principal functions: `krisp_fasta` and `krisp_vcf`. `Krisp_fasta` is used to infer diagnostic sites from whole genome assemblies based on shared unique k-mers. `Krisp_vcf` infers diagnostic sites from VCF files by analyzing a sliding window of variants and doing localized sequence inference. Each of these functions have a command line interface that is installed along with the package. `Krisp` is open source and available on GitHub with a user guide and test data (https://github.com/grunwaldlab/krisp) and on the Python PyPI package repository (https://pypi.org/project/krisp/).

### Krisp_fasta

The `krisp_fasta` command is designed to find all diagnostic regions differentiating one set of sequences from another. Sequences for both target and nontarget organisms are broken down into k-mers representing potential diagnostic regions and primer sites. These k-mers are filtered on the following criteria: 1) The presence of polymorphisms that distinguish the target group from the non-target group and 2) sequence conserved in the target group on either side of the polymorphic region where primers can be designed. `Krisp_fasta` accepts a FASTA file representing each sample. FASTA files can be passed to the command as an "ingroup" file or "outgroup" file, corresponding to target and non-target organisms. Results in comma-separated value (CSV) format are streamed to standard output or saved to a file. Optionally, a more human-readable alignment format can be saved to a text file as well.

The algorithm relies on extracting, sorting, and intersecting k-mers to find diagnostic regions. For each input FASTA file, all k-mers of length 'A' (short for amplicon) are extracted. Included in length 'A' are regions of length 'F' and 'R', corresponding to conserved regions where primers could be designed on the ends of the amplicon. 'F' and 'R' surround a diagnostic region of length 'D', such that A = F + D + R (Fig 1). K-mers of length 'A' are extracted and sorted by the sequence content in the 'F' and 'R' regions. Sorted k-mers for each input FASTA file are written to intermediate files in parallel, along with the name of the file they came from. Pairs of k-mer files are then read in tandem and combined into new files of sorted k-mers that include the names of the files each k-mer was observed in. Pairs of k-mer files are processed in parallel. This process is repeated with its own output, combining pairs of files, until a single file is left that includes the sorted k-mers for all input FASTA files. In this file, k-mers that share identical 'F' and 'R' regions, but differ in the 'D' region, are grouped together since they were sorted previously. Each of these groups of consecutive k-mers is referred to as an alignment.

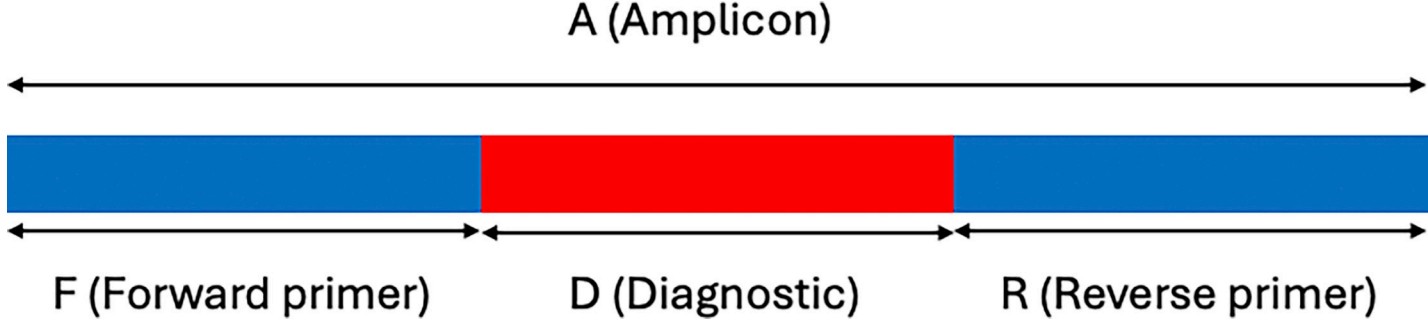

**Fig 1. Definition of amplicon, diagnostic, and primer regions for developing candidate CRISPR-Cas assays with `Krisp_fasta`.** The total amplicon length A is subdivided into non-overlapping regions of length F, D, and R, such that A = F + D + R, where F and R correspond to the forward and reverse primer regions, and D corresponds to the diagnostic region.

Alignments with a single k-mer present in all the target group files and not present in any of the non-target group files are considered diagnostic. Primer3 is then used to search for potential primer sites in these diagnostic k-mers. Alignments with diagnostic k-mers and predicted primers are formatted for output in two formats: a human-readable alignment format in a plain text file and a CSV file for further bioinformatic analysis. Both output formats contain the sequence of the diagnostic region and the output of Primer3 for the best primers found.

### Krisp_vcf

The `krisp_vcf` function looks for clusters of diagnostic variants, referred to here as diagnostic regions, flanked by regions without variants where primers can be designed. Diagnostic variants are those that are conserved and exclusive to the target group. Variants are read from a VCF file with an associated reference file in FASTA format. A CSV file must be supplied that encodes which samples in the VCF belong to which group. Diagnostic regions can be discovered for two or more groups with a single execution of `krisp_vcf`. VCF data can be streamed from standard input or read from a file. The output of `krisp_vcf` can be streamed to standard out or written to a file. Additionally, a log containing progress and error messages can be streamed to standard error or written to a file.

`Krisp_vcf` uses a sliding window analysis of consecutive variants for each group of diagnostic regions evaluated. First, each chromosome (i.e., each sequence in the reference FASTA) is broken up into chunks to enable parallel processing, unless VCF data is being streamed from standard input, in which case parallel processing is not possible. For each chunk, variants are read in order and filtered by number of reads, number of samples, genotype quality, and other quality control metrics. For each group, variants passing these filters are supplied to a series of first in, first out (FIFO) queues representing the upstream region where the reverse primer will be designed, the diagnostic region where the crRNA will be designed, and the downstream region where the forward primer will be designed. When a variant is added to any of these queues, the sequence length spanned by the variants is inferred and if this is longer than a set value, variants from the queue are moved to the next queue until the inferred sequence length is short enough. The diagnostic region queue holds the maximum number of variants that can fit in the length of a crRNA and the maximum length of the primer queues is determined by the maximum amplicon length. Once a variant is removed from all of the queues, it is removed from memory, so only variants present in the queues contribute to the RAM usage of the program. Once all the queues have been filled, each time a variant is added to the series of queues described above, the variants in the queues are subjected to a series of

tests to determine if they represent a diagnostic region. This effectively scans all possible regions in all samples. The following conditions must be met in order for the region to be considered diagnostic: 1. The diagnostic region queue must have enough diagnostic variants, 2. the variants in the diagnostic region queue must be conserved in the group of interest, 3. there must be enough conserved sequence in the group of interest in the primer queues to design primers, 4. A suitable primer pair must be found by Primer3 within the sequence inferred from the primer queues. These checks are ordered such that the most commonly failed and fastest to execute are done first in order to minimize processing time.

For regions passing all checks, the consensus sequence for the group of interest is inferred by applying the variants to the reference sequence and returned in the output. The output takes the form of a CSV file with columns for which group the region/primers is diagnostic for, the chromosome it occurs on, the coordinates of the reference genome where the primers and diagnostic region occur, the inferred sequence of the amplicon and surrounding sequence, and the Primer3 output. This format allows for additional downstream analysis with programming languages or spreadsheet programs, including giving users the sequence needed to design their own primers manually if desired. In addition to the CSV output, a human-readable text-based alignment output is provided that contains a multiple sequence alignment of the consensus sequence for each group being distinguished with annotations for the primer and crRNA locations. All Primer3 output is included in both the CSV and alignment-based formats.

## Performance evaluation

The performance of `krisp_fasta` was tested using a dataset of 12 assembled yeast genomes downloaded from NCBI: 6 genomes of baker's yeast (*Saccharomyces cerevisiae*), and 6 genomes of the closely related budding yeast (*Saccharomyces kudriavzevii*), each roughly 10–12 mega-bases in length (Table 1). The effect of the number of cores used on run time was tested on two computing systems, a desktop with a 3.20GHz 6-core Intel i7-8700 CPU and 32GB of RAM, and a computing cluster with a 2GHz 64-core AMD EPYC 7992 CPU and 4GB of RAM. For both systems, `krisp_fasta` was instructed to find all genomic regions which distinguish these two species with a diagnostic region of length 10 and conserved primer region of length 20. The effects of the number of samples, mean genome length, and the length of the amplicon (i.e., k-mer) on run time and RAM were also evaluated on a laptop computer with an Intel Core i7-10875H CPU @ 2.30GHz × 16 processor.

**Table 1. *Saccharomyces* genomes used to validate the `krisp` algorithm.**

| Species | Strain | Accession | Sequence length (Mbp) | Number scaffolds | Reference |
|---|---|---|---|---|---|
| *S. cerevisiae* | S288C | GCA_000146045.2 | 12.2 | 17 | [35] |
| *S. cerevisiae* | ySR127 | GCA_001051215.1 | 12.1 | 17 | [36] |
| *S. cerevisiae* | BY4742 | GCA_003086655.1 | 12.2 | 16 | [37] |
| *S. cerevisiae* | KSD-Yc | GCA_003709285.1 | 12.0 | 16 | |
| *S. cerevisiae* | ySR128 | GCA_004328465.1 | 12.1 | 17 | [38] |
| *S. cerevisiae* | IMF17 | GCA_018219195.1 | 12.4 | 19 | [39] |
| *S. kudriavzevii* | IFO 1802 | GCA_000167075.2 | 11.2 | 2,054 | [40] |
| *S. kudriavzevii* | ZP591 | GCA_000257045.1 | 10.7 | 1,814 | [41] |
| *S. kudriavzevii* | IFO1803 | GCA_000257065.1 | 10.4 | 1,798 | [41] |
| *S. kudriavzevii* | IFO10990 | GCA_000257105.1 | 10.5 | 1,817 | [41] |
| *S. kudriavzevii* | CR85 | GCA_003327635.1 | 11.9 | 17 | |
| *S. kudriavzevii* | CR85 | GCA_900682695.1 | 11.6 | 16 | |

The performance of `krisp_vcf` was evaluated using a 20Gb VCF file containing 174,743,308 variants for 656 *Phytophthora ramorum* samples, grouped by clonal lineage. The 57.5 Mb reference genome of strain PR-102_v3.1 was used [29]. These tests were conducted on a laptop computer with an Intel Core i7-10875H CPU @ 2.30GHz × 16 processor. The effect of the number of cores on execution time was evaluated on a subset of this dataset. The effects of number of samples, the number of groups compared, and the number of variants were evaluated using the same dataset and computer by subsetting the dataset as needed.

## Validating `krisp_fasta` and `krisp_vcf`

The output of `krisp_fasta` and `krisp_vcf` were validated in the lab by using both to design a SHERLOCK diagnostic assay to differentiate the plant pathogen *Phytophthora ramorum* from closely related *Phytophtora* species [30]. *P. ramorum* is a destructive pathogen killing a variety of forest trees on the west coast of the United States [31]. It is thought to be spread primarily through trade of nursery stock and regulations are in place to stop the selling of infected plants. Diagnostic assays are needed to quickly and cheaply identify *P. ramorum* infections in order to prevent its spread and comply with regulations. For `krisp_fasta`, whole genome sequences from 5 *P. ramorum* isolates (PR-102, PR-15-019, PR-18-069, PR-18-108, and PR-18-126) were used to represent the diversity within *P. ramorum* and genome sequences of *P. brassicae*, *P. cryptogea*, *P. foliorum*, *P. hibernalis*, *P. lateralis*, and *P. syringae* were used to represent non-target *Phytophthora* species. In addition, a version of the *P. ramorum* genome PR-102 with variable regions masked with Ns was included. The masking was done by analyzing a VCF file containing published variants of *P. ramorum* samples [32–34] and converting any position homozygous for the alternative allele to N. This allowed for the incorporation of data from many samples that do not have assembled genomes available. For `krisp_vcf`, a VCF file containing variants of *P. ramorum* samples from available genomes [32–34] and the PR-102_v3.1 reference sequence was used as input [29]. A promising diagnostic site was selected from the many options produced by `krisp_fasta` and checked against analogous VCF data using `krisp_vcf`.

Briefly, we evaluated the ability to distinguish *P. ramorum* from other *Phytophthora* taxa with a SHERLOCK assay. SHERLOCK is a two-step assay starting with RPA of target DNA. The forward Primer for RPA included a T7 promoter region at the 5' end that, when combined with the reverse primer, produced a 117 bp amplicon in the target region (Forward primer: GAAATTAATACGACTCACTATAGGGTGCATTTTCGACAAATTCGAGTGCGGGGT CAG, Reverse primer: ATCGAAATATCGGCGCGTCCATAACGGTCATA). Amplification reactions were prepared according to Kellner et al. [12], using a master mix that included 10μM primers, water, and TwistAmp rehydration buffer (Twistdx, Maidenhead, UK). This master mix was used to rehydrate the TwistAmp polymerase followed by the addition of 280mM of magnesium acetate. Ten microliter aliquots of the reconstituted RPA were added to PCR tubes along with 1μl of template and placed in a thermocycler set at 37C for 30 min. The crRNA 5'-UUAUCCGAGCCCGUGAUGAAGUUGUUGC-3' was designed for the Protein phosphatase 2 (PP2A) regulatory subunit B locus. The 5th base was modified from an 'A' to a 'C' to introduce 1 mismatch into the crRNA-target alignment. For *P. ramorum*, this meant that the crRNA and target have a single mismatch. For other *Phytophthora* species, there were at least two mismatches and thus no collateral cleavage should occur. Adding the conserved DR region for LwaCas13a results in 5'-ACUACCCCAAAAACGAAGGGGACUAAAACUU AUCCGAGCCCGUGAUGAAGUUGUUGC-3'. For detection, a master mix was prepared containing ultrapure water, 20mM HEPES pH 6.8, 9.5mM MgCl, 1mM rNTP solution mix, 6.7μg LwaCas13a (MCLAB, San Francisco, CA or Genscript, Piscataway, NJ), 40U Murine

RNase inhibitor (New England Biolabs, Ipswitch, MA), 2.5U T7 RNA polymerase (New England Biolabs), 10ng/μl CRISPR guide RNA (IDT, Coralville, IA), and 0.13μM RNaseAlert v2 (Thermo Fisher, Waltham, MA) per reaction. Four replicates of each sample were aliquoted into a 384 well plate (20μl per well), centrifuged and immediately placed in a fluorescent plate reader (Tecan, Switzerland) preheated to 37C. Fluorescence (490/520nm) was recorded for 3h at 5min intervals. Positive controls included known samples of *P. ramorum* from clonal lineages NA1, NA2, EU1 and EU2, whereas negative controls included samples of *Phytophthora* species including *P. cinnamomi*, *P. foliorum*, *P. lateralis*, and *P. plurivora* as well as water. Background subtracted fluorescence was graphed over time.

## Results

`Krisp` is a Python package for finding candidate regions for the development of CRISPR-dx diagnostic assays. `Krisp` can analyze whole genome sequence data in the form of FASTA files or variant data in the form of a VCF file with an associated reference in FASTA format. Primer3 is used to screen potential diagnostic regions for suitable primer binding sites. Diagnostic regions are output in the form of a CSV file or human-readable plain text alignments. `Krisp` has been optimized to minimize RAM use and can run in parallel, allowing large datasets to be processed on personal computer in a matter of hours.

### Performance

On both the desktop and the cluster, `krisp_fasta` ran about twice as fast when 6 cores were used compared to using a single core. Further increasing the number of cores on the desktop computer provided no increase in speed. When run on the computing cluster, speed increased through 12 cores (Fig A in S1 Text). In terms of absolute time, the desktop computer took ~40 minutes to complete processing the test dataset of 12 yeast genomes with 1 core, which decreased to ~20 minutes when using 6 cores. The computing cluster utilizes a slower CPU and took ~80 minutes to complete with 1 core, which decreased to ~25 minutes with 12 cores (Fig A in S1 Text). The `krisp_fasta` algorithm utilizes very little memory during execution, since it uses intermediate files instead of RAM for memory-intensive steps. Although RAM usage appears to correlate with input sequence length (Fig 2), this is due to the Linux `sort` utility taking advantage of excess RAM to operate faster. If less RAM were available the program would still run. `Krisp_vcf` achieves a low memory footprint as well by only loading variants present in a sliding window. This allows `krisp` to run on computing systems with as little as 4GB of RAM, making it suitable for laptops, desktops, and similar personal computers. The effects of the number of samples, mean genome length, and the length of the amplicon (i.e. k-mer) on run time and RAM were confirmed to be linear as expected (Fig 2).

`Krisp_vcf` completed analysis of an entire 20Gb VCF file containing 13 million variants for 656 samples in 24 minutes when using 16 cores on a laptop. The effect of the number of cores used on the execution time was tested using a subset of the same data. Increasing the number of cores from 1 to 4 decreased execution time nearly 4-fold, but additional cores had little effect (Fig B in S1 Text). The effects of number of samples, the number of groups compared, and the number of variants on execution time was confirmed to be approximately linear, although the trend was somewhat noisy (Fig 3). RAM usage was consistently around 170Mb regardless of dataset scale.

### Laboratory validation

The outputs of both `krisp_vcf` and `krisp_fasta` were used to design a SHERLOCK proof-of-concept assay (Fig 4A). The assay was built to distinguish the sudden oak death

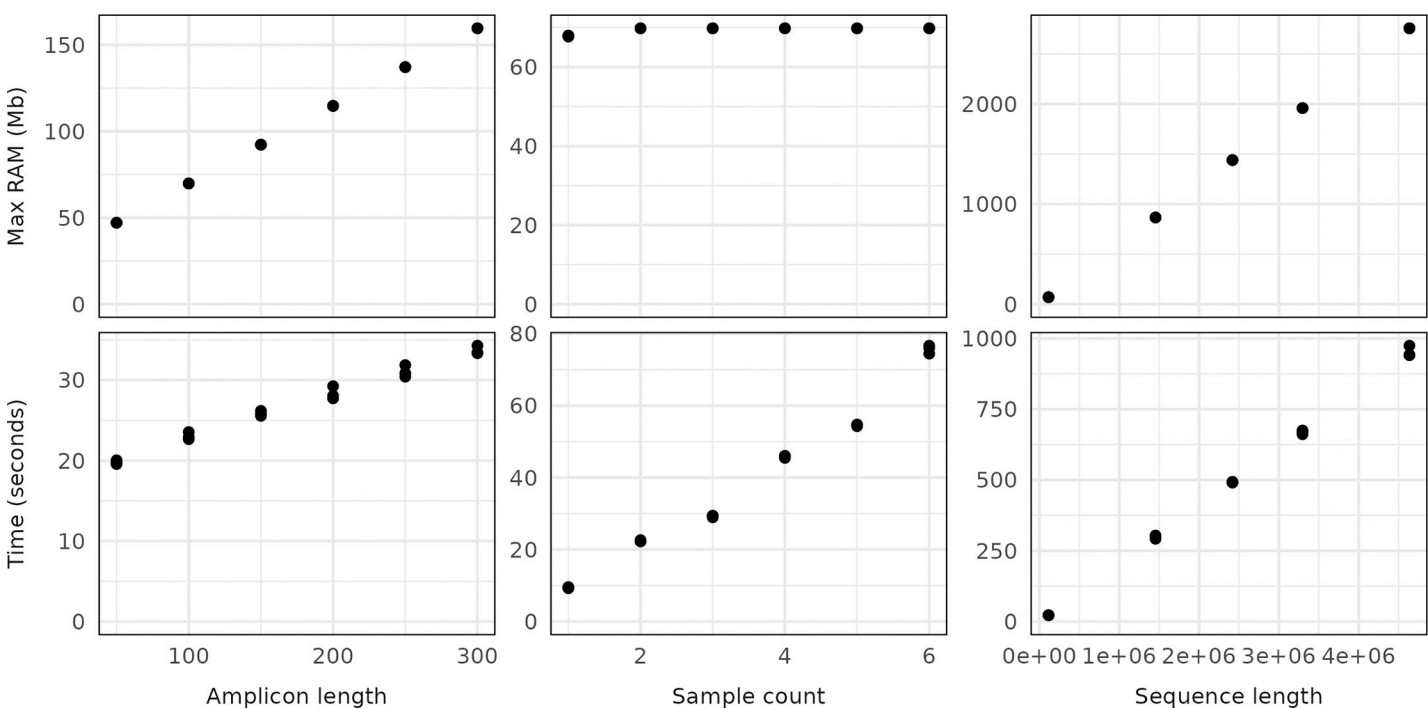

**Fig 2. The effects of amplicon (k-mer) length, sample count, and mean genome length on the execution time and maximum RAM usage of krisp_fasta.** For each column, variables not being tested were held constant with the following values: amplicon length of 100bp, sample count of 2, and a mean sequence length of 69,808 (the length of the first chromosome in the test dataset). Testing was done a laptop computer with an Intel Core i7-10875H CPU @ 2.30GHz × 16 using a single core.

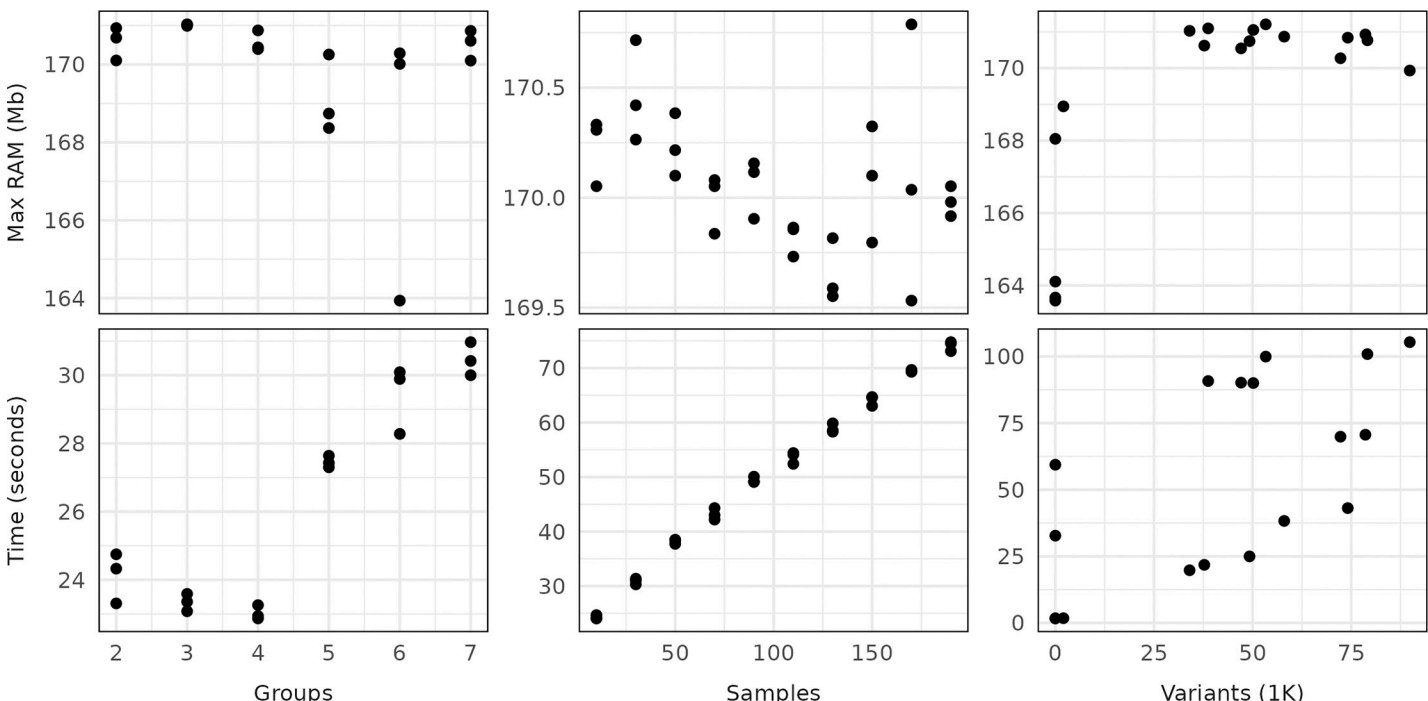

**Fig 3. The effects of variant count, sample count, and number of groups being distinguished on the execution time and maximum RAM usage of krisp_vcf.** For the sample count and group count tests, variant count was held constant at the number of variants that occur in the first 500,000bp of the first chromosome. For the variant count and group count tests, sample count was held constant at a total of 6 and 24 samples respectively. For the variant count and sample count tests, the number of groups was held constant at 2. Testing was done a laptop computer with an Intel Core i7-10875H CPU @ 2.30GHz × 16 using a single core.

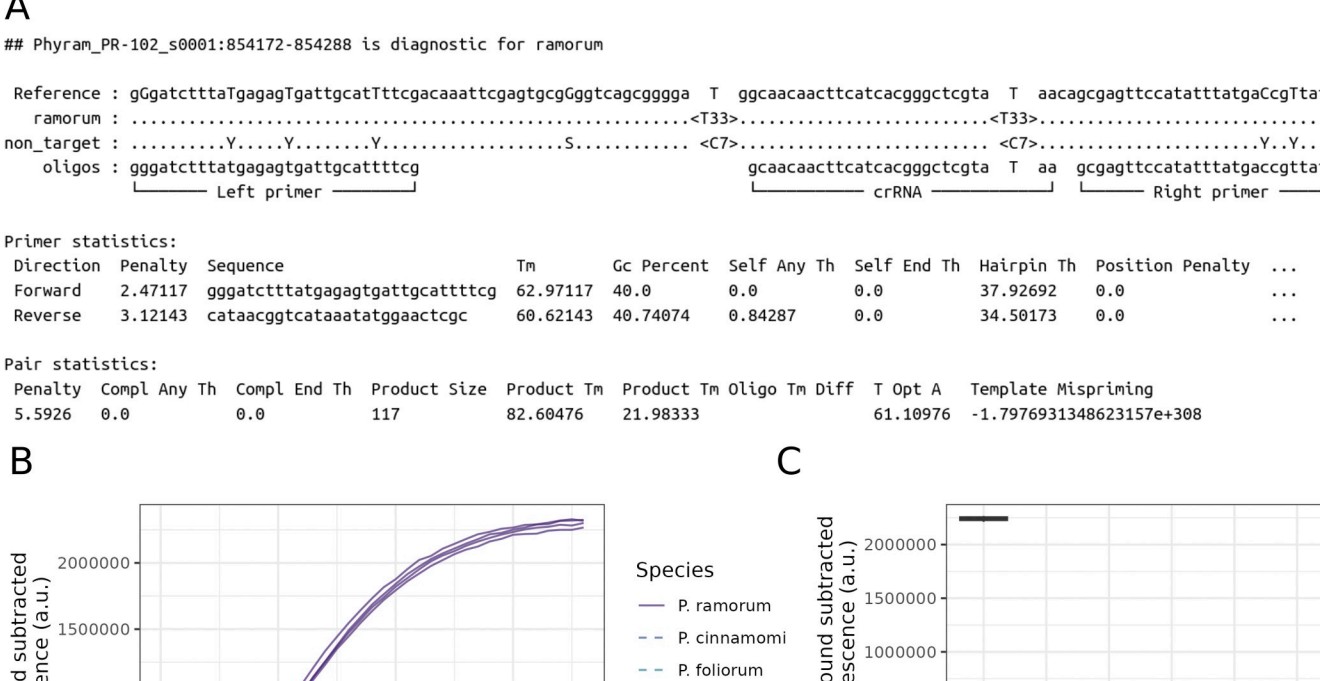

**Fig 4.** Proof-of-concept assay predicted by both `krisp_vcf` and `krisp_fasta` to create a SHERLOCK CRISPR-dx assay for the Protein phosphatase 2 (PP2A) regulatory subunit B locus to detect the species *Phytophthora ramorum*. **A:** Abbreviated output from `krisp_vcf` showing predicted primers and diagnostic crRNA region for detection of *P. ramorum*. Note, that the primers we initially used for validation were different, but the crRNA locus is identical. **B:** *P. ramorum* clonal lineages NA1, NA2, EU1, EU2 as well as Asian strains (purple solid lines) could be detected while non-target species *Phytophthora cinnamomi*, *P. foliorum*, *P. lateralis*, and *P. plurivora* (shown in various colors as dashed lines) did not amplify. **C:** Distributions of background subtracted fluorescence at 150 minutes for *P. ramorum* lineages and non-target species and controls.

pathogen *Phytophthora ramorum* from other closely related *Phytophthora* species. Our results suggest that both programs work as expected. All *P. ramorum* variants NA1, NA2, EU1 and EU2 resulted in fluorescence whereas negative controls, including 6 other *Phytophthora* species, did not (Fig 4B and 4C).

## Discussion

The ability to rapidly develop diagnostic tests to distinguish closely related organisms would be a useful tool for tracking the spread of emerging pathogens and organizing an effective response. CRISPR-dx technology has the potential to produce highly specific, sensitive, and inexpensive tests that can be administered with minimal specialized equipment or expertise. Quickly identifying candidate sequences to design diagnostic crRNAs and primers that can be tested in the laboratory will decrease the time it takes to deploy new assays. Krisp can be used to find candidate diagnostic regions in which to design CRISPR-dx or amplification-based assays to differentiate one group of organisms from another for any species for which assembled sequences or variant data is available. Krisp has been carefully optimized to

handle large datasets of commonly available data formats and leverage parallel computing to rapidly produce actionable results.

To demonstrate the usage of `krisp_fasta`, below we show how it can be used on the yeast dataset described above to locate regions where a CRISPR-dx assay could be designed (Table 1). We aim to find diagnostic regions which differentiate the 6 baker's yeast genomes from the 6 budding yeast genomes. More specifically, we instruct `krisp` to find all genomic regions in which a diagnostic region of length 10 is flanked by conserved primer regions of 50 nucleotides in both directions:

```
krisp_fasta cerevisiae/*.fna.gz -outgroup kudriavzevii/*.fna.
gz -conserved 50 -diagnostic 10 -out_align align.txt -out_csv
out.csv -primer3
```

In this example, `krisp_fasta` was able to find 120 candidate regions which fit the criteria. Below is one of those regions (parts of the output are abbreviated with '...' for visualization):

```
TGCAAGTTAATTGGAACGGAAGCACC...TTGTCAACTTGAAC...AGATGAAATCTTACCTTCTTGACCCTT: SC1;SC2;SC3;SC4;SC5;SC6
TGCAAGTTAATTGGAACGGAAGCACC...TTGTCAACCTGGAC...AGATGAAATCTTACCTTCTTGACCCTT: SK1;SK2;SK3;SK4;SK5;SK6
└──────────Forward──────────┘ ... {──#-#} ... └──────────Reverse──────────┘

Primer statistics:
 Direction  Penalty   Sequence                    Tm          Gc Percent  Self Any Th  Self End Th  ...
 Forward    7.22384   TGCAAGTTAATTGGAACGGAAGCAC   62.72384    44.0        0.0          0.0          ...
 Reverse    6.91895   AGGGTCAAGAAGGTAAGATTTCATC   58.58105    40.0        0.0          0.0          ...
Pair statistics:
 Penalty    Compl Any Th  Compl End Th  Product Size  Product Tm  Product Tm Oligo Tm Diff       ...
 14.14278   0.0          0.0           109           79.88148    21.30042                       ...
```

Candidate regions can be output in an alignment format where each unique sequence is stacked vertically and the corresponding genome files are listed on the right. In this example, there are two sequences and twelve genome files with names starting with 'SC' or 'SK', corresponding to the genome files for *S. cerevisiae* and *S. kudriavzevii*, respectively. The first sequence is associated with six genome files, all of which correspond to *S. cerevisiae*, implying that this sequence is conserved across the entire ingroup, whereas the second sequence was found exclusively in the outgroup *S. kudriavzevii*. Since only unique sequences are displayed, it is common to have multiple files associated with a single sequence in an alignment. When a sequence is found multiple times in a genome file, the number of times it occurred is appended to the genome file name in the format of '(n)'. For example, 'SC1(2)' would imply that this sequence was detected twice in the genome file SC1. Near-identical sequence matches would be listed as separate sequence entries, assuming the only differences are within the diagnostic region. The last line of the alignment shows a summary, where '{}' denotes the boundaries of the diagnostic region, '-' a conserved position, '*' a non-diagnostic SNP, and '#' a diagnostic SNP with respect to the ingroup. In this case we see that the diagnostic region contains two diagnostic SNP's, a 'T-C' and 'A-G' difference, and zero non-diagnostic SNP's. The alignment is annotated with the locations of the best primers found by Primer3. The full output of Primer3 for these primers is printed in a tabular format below the alignment for convenient manual inspection.

In the case of VCF input, `krisp_vcf` can be used in a similar way to `krisp_fasta`. For this example, we will use a subset of the of *P. ramorum* VCF data described above that is included in the package as a test dataset. We search for regions that distinguish each of three clonal lineages of *P. ramorum* from all other lineages with the following command:

```
krisp_vcf metadata.csv reference.fasta -vcf variants.vcf
-groups NA1 NA2 EU1 -out_align alignments.txt -gc_clamp 2
```

The "metadata.csv" file contains a table with two columns: each sample's name as it appears in the VCF file and what group (lineage in this case) that the sample belongs to. The "—groups" option defines which groups assays should be designed for. The "—gc_clamp 2" option is one of many Primer3 settings that can be changed; this option instructs Primer3 to only consider primers with at least 2 bases that are G or C on the 3' end of both primers. The command above produces the following alignment, among many others (parts of the output are shortened with '. . .' for visualization):

```
## Phyram_PR-102_s0001:209731-209856 is diagnostic for NA2
Reference: gtcccgtcaccgtatatatgtactaaacgca...gtgggtagcatactgacgacgagaagt  C
agt...ctggttaggacagttaaatgtaccGagag
     EU1: ....................................................  C9 ...................R...
     NA1: ....................................................  C6 .............................
     NA2: ...................................................<T11>.............................
  oligos: gtcccgtcaccgtatatatgtactaaacg ... gggtagcatactgacgacgagaagt  T  ag ... ggttaggacagttaaatgtaccgagag
         └────── Left primer ──────┘ ... └────── crRNA ──────┘ ... └────── Right primer ──────┘
Primer statistics:
 Direction  Penalty  Sequence                        Tm        Gc Percent  Self Any Th  Self End Th  Hairpin Th
 Forward    3.42431  gtcccgtcaccgtatatatgtactaaacg   62.92431  44.82759    7.87208      10.60241      0.0      ...
 Reverse    3.60529  ctctcggtacatttaactgtcctaacc     61.10529  44.44444    1.18023      3.98189       0.0      ...
Pair statistics:
 Penalty  Compl Any Th  Compl End Th  Product Size  Product Tm  Product Tm Oligo Tm Diff  T Opt A            ...
 7.0296   2.27423       0.0           126           83.22136    22.11607                  61.68654          ...
```

In this format, columns in the alignment with a diagnostic variant have the alleles and their counts listed in the form of the allele sequence followed by the number of samples that have that allele. For example, the "C9" in the above output means that all 9 of the samples assigned to the EU1 lineage have C in that position. If multiple alleles were present, they would be listed in series (e.g., C5T1). Diagnostic alleles are highlighted with angle brackets. For example, in the above output "<T11>" means that all 11 samples of NA2 have T at that position and none of the samples from other lineages do. Variants not located in diagnostic columns are indicated by capital letters, potentially using IUPAC ambiguity codes if a group has multiple alleles at a given position. Like `krisp_fasta`, the Primer3 output for the best primers is presented below the alignment.

`Krisp` is designed to leverage massive datasets of whole genomes to find diagnostic regions suitable for any amplification-based assay, but this efficiency comes with some drawbacks that could make other software more appropriate in some situations. While other software for CRISPR-dx design, such as CaSilico [26] and PrimedSerlock [27], can estimate optimal crRNA for specific types of Cas enzymes or assays, `krisp` is more generalized in order to be useful for any amplification-based diagnostic assay. Similarly, the primer design considerations change depending on assay type. `Krisp` uses Primer3 to find primers because it runs quickly with minimal resources, allowing for huge numbers of potential sites to be checked quickly. However, Primer3 cannot find primers for some isothermic amplification techniques like LAMP. Given these limitations, users might have to modify the suggested location of crRNAs or primers depending on the assay type, but `krisp` is designed to make this process as easy as possible by providing the sequence surrounding each potential diagnostic region. `Krisp` does not rely on multiple sequence alignments, which are difficult to produce reliably and quickly for many genome-scale sequences, allowing it to analyze massive datasets,

but the alternative methods employed have their own considerations. `Krisp_vcf` uses VCF data as input, which relies on mapping reads to a single reference. If the input sequences are too diverse to map to a single reference, then false negative results are possible in the unmapped areas. `Krisp_fasta`'s reliance on k-mers mean that it does not handle indels, which could also lead to false negatives.

We hope that the `krisp` algorithm will facilitate development of novel diagnostic assays. A complete step-by-step design of a CRISPR-dx assay is beyond the scope of this paper; here, we provide a starting point for more detailed and customized analyses. The designing of a typical CRISPR-dx assay can be broken down into two steps, corresponding to primer-based amplification and CRISPR-Cas based recognition via a crRNA. `Krisp` is designed to provide possible candidates for both, but final selection of a crRNA and primers will depend on many factors, including the Cas enzyme and type of amplification (e.g., RPA or LAMP) used. For example, the LwCas13a enzyme used in SHERLOCK can only achieve single base pair resolution when the SNP is in the 3$^{rd}$ position from the 3' end and an artificial mismatch is added nearby. `Krisp_vcf` takes this into account by positioning the diagnostic SNPs at a location specified by the user. We hope to add additional functionality to `krisp` in the future to consider such technique-specific details to further decrease the workload needed to design CRISPR-dx or other diagnostic assays. Finally, `krisp` can be used beyond CRIPSR-dx applications to search genome or VCF data for any combination of user specified diagnostic probes and/or primers distinguishing target and non-target groups.

## Conclusions

We developed the computational tool `krisp` to identify genetic regions suitable for development of DNA or RNA based diagnostic assays. `Krisp` scans whole genome sequence data for target and non-target groups to identify diagnostic regions based on DNA or RNA sequences. This tool can be used for any organism at a variety of taxonomic levels (e.g. genus, species, sub-species) for which assembled whole genome sequences or VCF data are available representing the genetic diversity of the samples to be distinguished. `Krisp`'s primary benefits over other approaches are speed, scale, and automation. While it is possible to manually identify diagnostic regions from whole genome sequences alignments, VCFs, or BAM pileups using programs with graphical user interfaces, this usually requires a priori knowledge of which genes might be useful in this regard to be done within a practical amount of time and effort. `Krisp` does not require the user to know which part of the genome to look at because its efficiency allows the entire genome to be searched. Additionally, `krisp` does not rely on multiple sequence alignments like most similar programs do, since multiple sequence alignments are often unreliable and difficult to produce when there are large scale genomic rearrangements and many genomes to align. This computational tool has been validated in silico and experimentally with a species-specific SHERLOCK assay. `Krisp` is released open source under the MIT license and available on GitHub (https://github.com/grunwaldlab/krisp) and the Python PyPI package repository (https://pypi.org/project/krisp/). A user guide with examples is provided on the GitHub repository.

## Supporting information

**S1 Text. Supplementary Materials.**
(DOCX)

## Author Contributions

**Conceptualization:** Zachary S. L. Foster, Andrew S. Tupper, Niklaus J. Grünwald.

**Data curation:** Caroline M. Press.

**Formal analysis:** Zachary S. L. Foster, Andrew S. Tupper, Caroline M. Press, Niklaus J. Grünwald.

**Funding acquisition:** Niklaus J. Grünwald.

**Investigation:** Zachary S. L. Foster, Andrew S. Tupper, Caroline M. Press, Niklaus J. Grünwald.

**Methodology:** Zachary S. L. Foster, Andrew S. Tupper, Caroline M. Press, Niklaus J. Grünwald.

**Project administration:** Niklaus J. Grünwald.

**Resources:** Caroline M. Press, Niklaus J. Grünwald.

**Software:** Zachary S. L. Foster, Andrew S. Tupper, Niklaus J. Grünwald.

**Supervision:** Niklaus J. Grünwald.

**Validation:** Caroline M. Press, Niklaus J. Grünwald.

**Visualization:** Zachary S. L. Foster, Andrew S. Tupper, Caroline M. Press, Niklaus J. Grünwald.

**Writing – original draft:** Zachary S. L. Foster, Andrew S. Tupper, Caroline M. Press, Niklaus J. Grünwald.

**Writing – review & editing:** Niklaus J. Grünwald.

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
