## [Decision Letter · Decision Letter 0]

12 Feb 2024

Dear Dr. Grunwald,

Thank you very much for submitting your manuscript "Krisp: A python package for designing CRISPR and amplification-based diagnostic assays from whole genome data" for consideration at PLOS Computational Biology.

As with all papers reviewed by the journal, your manuscript was reviewed by members of the editorial board and by several independent reviewers. In light of the reviews (below this email), we would like to invite the resubmission of a significantly-revised version that takes into account the reviewers' comments. Particularly important is the request to expand on the description of the field and to evaluate the performance of the method in light of existing approaches.

We cannot make any decision about publication until we have seen the revised manuscript and your response to the reviewers' comments. Your revised manuscript is also likely to be sent to reviewers for further evaluation.

Sincerely,

Yana Bromberg

Guest Editor

PLOS Computational Biology

Ilya Ioshikhes

Section Editor

PLOS Computational Biology

Reviewer's Responses to Questions

**Comments to the Authors:**

Reviewer #1: The paper by Foster, et al. et al. describes a Python library, Krisp, that allows detection of target regions and design of primers for CRISPR applications that are able to discriminate between different strains (e.g.). The authors describe the functioning of the package, and how to use it. They cover benchmarking of the speed/CPU usage, and provide a basic example of application to sequences. The paper is written well but I have the following comments for improvement.

1. Though the tool looks useful and the paper is well written, it is very light on any biological demonstration or interpretation. It is useful to show the P. ramorum data – and that’s great, but this doesn’t really communicate the biological impact of the approach (what would the alternative have been, e.g., in terms of primer design). The examples given are fine to given an idea of how the tool functions – but it is difficult to assess how useful the tool will be, or what kind of biological insights it might provide.

2. There is a lot of benchmarking on the computational performance of the tool that is probably not necessary in the main text. I’ve included some suggestions below.

3. Figures 2 and 3 are of limited value given that Figure 2 is only comparing the algorithm’s performance against itself, and Figure 3 can be easily summarized as text (not sure the figure adds much really).

4. Figure 4 (bottom panel) is a bit confusing. I believe that the plot is supposed to be showing the difference between control (dashed lines) and experiment (solid lines) – but the plot looks like there are a lot more solid lines (maybe the clonal lineages) than dashed lines. I think this would be better conveyed as a bar graph (e.g.) showing the differences of all the strains at 150 minutes. The time doesn’t seem to add a lot.

5. Line 334, “Expected Performance” – this section does not need to be in the main text, and I feel could be moved to supplement. The expected performance is somewhat interesting, but only in light of the realized performance (in the next section).

Minor comments.

6. Line 22, SHERLOCK has not been defined – so it’s an odd thing to talk about in the Abstract.

7. Line 140, WGS should be defined here.

8. Python should be capitalized throughout.

9. Line 258, RAM should be capitalized

10. Line 267, ‘P. ramorum’ has not been spelled out previously.

11. Line 372, “On both computing systems” – both of which systems?

12. Line 372, “twice as fast when 6 cores were used.” – twice as fast as what?

13. Line 374, “hardware limits being reached.” – while true, not very helpful. My guess would be IO limitations? Limitations on threading relative to the algorithm?

14. Line 376, “processing the test dataset” – what was the test dataset?

15. Line 393, “likely due to limitations of the laptop cooling system” – this seems… odd. There seem to be more likely explanations. I’m not sure any explanation without evidence is needed here.

Software

1. Install from PyPi worked well

2. The Github page Requirements indicates that Python 3.6 or better is needed but it seems that the required version is Python 3.9 or better.

3. I was not able to get krispr_fasta to function with my install (latest install of Python 3.12, but other older versions still on my machine) as it seemed to be calling my Python 3.8 (which doesn’t work – see point 2, above). Though I’m sure the fix is probably relatively simple I was unable to figure it out to provide any further review of the software for the paper in a timely manner (and I’m sure others may experience similar issues).

Reviewer #2: The authors have developed a Python package, Krisp, capable of detecting variations common among sequences of interest. The tool provides a range of additional features, making it versatile for multiple applications and readily available for installation via Pip. The manuscript and GitHub repo are descriptive, with detailed information about the tool. There are a few comments to improve the manuscript and usability of the tool:

1) There are several existing computational tools to aid CRISPR experiments. In the introduction, The authors need to summarize the current state of computational tools in the field and the importance of Krisp, a new computational tool. How does Krisp stand out or complement them? 

2) Currently, Krisp is restricted to searching for a fixed variation length among a specified length of conserved regions. However, one could improve Krisp's versatility in diverse applications and biological data by facilitating a search of a range of lengths (e.g., 3-5 bp ). 

3) What are the drawbacks/limitations of the tool?

Minor:

3) Can Krisp capture common variants among a set of distantly related species with deletions and insertions? One or two examples could help the readers.

4) How does the performance scale with the number of genomes or the sequence set size? The figures on scaling to the number of cores are of little use.

5) Is the pip version of your tool the same as the current GitHub version? 

6) What are the required columns in Krisp's VCF file? Do you know if the tool works with mutated VCF file formats?

Reviewer #3: This manuscript overall does indeed describe a reasonable approach to the goal to "find differentiating diagnostic sequences/primers for sequences of interest, in a memory efficient manner". However, this is somewhat at odds with the title and parts of the abstract, since finding primers using Primer3 is not the same thing as “A python package for designing CRISPR and amplification-based diagnostic

assays from whole genome data”. For example, the described work represents what is really a front end and workflow for whole-genome primer design that depends on Primer3, not a complete package, and it does not design CRISPR guide RNAs, just primers. Having said that, overall, the work on a primer design front end and workflow is appropriate and thorough. Note the contrast here with the conclusion, that they “identify genetic regions suitable for development of DNA or RNA based diagnostic assays” – this is very different from what the they say in the title and abstract.

The "chunking"/parallelization routines that the authors implemented, and demonstration of memory efficiency and discussion of performance with number of cores etc, are useful and not commonly implemented in this area.

However the reliance on Primer3, which is rather old software with a number of known issues (for example, oversensitivity to hairpin loop structures, and inaccurate Tm calculations, at least in the versions known to this reviewer) does reduce the potential impact of the work.

Fig. 4 was also refreshing in that the authors demonstrated real-world biological performance of their method, although this demo seemed to lack needed biological context. This figure demonstrates utility in the cross-species analysis presented, but the use case for cross-species primer design is unclear, and the application seems unusual and highly specific. A better rationalization could be made for this choice of proof of concept experiment.

All in all, the work provides a useful platform that is useful in conjunction with other tools/experiments in a larger workflow. To be suitable for publication, the title and abstract need to better reflect the work that was actually done, the performance of Primer3 needs to be discussed, and preferably the code described needs to accommodate alternative primer design software.

**Have the authors made all data and (if applicable) computational code underlying the findings in their manuscript fully available?**

Reviewer #1: Yes

Reviewer #2: Yes

Reviewer #3: Yes

PLOS authors have the option to publish the peer review history of their article (what does this mean?). If published, this will include your full peer review and any attached files.

Reviewer #1: No

Reviewer #2: No

Reviewer #3: No
---

## [Decision Letter · Decision Letter 1]

6 May 2024

Dear Dr. Grunwald,

We are pleased to inform you that your manuscript 'Krisp: A Python package to aid in the design of CRISPR and amplification-based diagnostic assays from whole genome sequencing data' has been provisionally accepted for publication in PLOS Computational Biology.

Best regards,

Yana Bromberg

Guest Editor

PLOS Computational Biology

Ilya Ioshikhes

Section Editor

PLOS Computational Biology

Please make sure to address the remaining comments regarding figures and missing explanations

Reviewer's Responses to Questions

**Comments to the Authors:**

Reviewer #1: The authors have addressed all my previous comments. I have also now successfully installed and tested the package.

Reviewer #2: The authors have addressed all my comments in the revised manuscript. However, the manuscript could make use of careful proofreading.

I suggest the authors to format the newly added figures 2&3 - to address the varying scales & units across subfigures, b) typos in the captions and c) the missing explanation for the chosen range of parameters (sample count, sequence length, etc).

Reviewer #3: The authors have appropriately addressed my comments. I have no further suggestions. Thank you for taking my suggestions seriously.

**Have the authors made all data and (if applicable) computational code underlying the findings in their manuscript fully available?**

Reviewer #1: Yes

Reviewer #2: Yes

Reviewer #3: Yes

PLOS authors have the option to publish the peer review history of their article (what does this mean?). If published, this will include your full peer review and any attached files.

Reviewer #1: No

Reviewer #2: No

Reviewer #3: No

---

## [Editor Report · Acceptance letter]

13 May 2024

PCOMPBIOL-D-24-00012R1 

Krisp: A Python package to aid in the design of CRISPR and amplification-based diagnostic assays from whole genome sequencing data

Dear Dr Grunwald,

I am pleased to inform you that your manuscript has been formally accepted for publication in PLOS Computational Biology. Your manuscript is now with our production department and you will be notified of the publication date in due course.

With kind regards,

Anita Estes
